# Methane Emissions from Ruminants in Australia: Mitigation Potential and Applicability of Mitigation Strategies

**DOI:** 10.3390/ani11040951

**Published:** 2021-03-29

**Authors:** John L. Black, Thomas M. Davison, Ilona Box

**Affiliations:** 1John L Black Consulting, Warrimoo, NSW 2774, Australia; 2Livestock Productivity Partnership, University of New England, Armidale, NSW 2351, Australia; thomas.davison@une.edu.au; 3Ilona Box Consulting, Warrimoo, NSW 2774, Australia; ilonabox8@gmail.com

**Keywords:** enteric methane, methane mitigation, genetic selection, vaccination, grape marc, nitrate, biochar, 3-nitrooxypropanol, *Asparagopsis*, rumen microbe manipulation

## Abstract

**Simple Summary:**

Methane is a potent greenhouse gas. It is 80-times more effective at heating the earth than carbon dioxide over the first 20 years following release into the atmosphere. Ruminant animals have diverse microbial populations in their stomachs that employ anaerobic fermentation to digest feed. Methane is belched into the atmosphere as a by-product of the digestive process. This gut, or enteric methane, primarily from cattle, but also sheep and goats, contributes 30% of the methane released into the earth’s atmosphere each day, and is more than any other single methane source. A major reduction in methane emissions from ruminants is crucial to preserve ecosystems on the planet. Various strategies to reduce enteric methane emissions in farm operations are reviewed to quantify their mitigation potential, determine their impact on animal productivity and likelihood of adoption. Two feed supplements, a commercial product, 3-NOP (Bovaer^®^), and the seaweed, *Asparagopsis*, can reduce methane emissions by 40+% and 90%, respectively, with associated increases in animal productivity and no adverse effects on animal health or product quality. The rumen microbial population can also be changed to provide long-term intergenerational reduction in methane emissions, if treated herds remain isolated from non-treated animals.

**Abstract:**

Anthropomorphic greenhouse gases are raising the temperature of the earth and threatening ecosystems. Since 1950 atmospheric carbon dioxide has increased 28%, while methane has increased 70%. Methane, over the first 20 years after release, has 80-times more warming potential as a greenhouse gas than carbon dioxide. Enteric methane from microbial fermentation of plant material by ruminants contributes 30% of methane released into the atmosphere, which is more than any other single source. Numerous strategies were reviewed to quantify their methane mitigation potential, their impact on animal productivity and their likelihood of adoption. The supplements, 3-nitrooxypropanol and the seaweed, *Asparagopsis*, reduced methane emissions by 40+% and 90%, respectively, with increases in animal productivity and small effects on animal health or product quality. Manipulation of the rumen microbial population can potentially provide intergenerational reduction in methane emissions, if treated animals remain isolated. Genetic selection, vaccination, grape marc, nitrate or biochar reduced methane emissions by 10% or less. Best management practices and cattle browsing legumes, *Desmanthus* or *Leucaena* species, result in small levels of methane mitigation and improved animal productivity. Feeding large amounts daily of ground wheat reduced methane emissions by around 35% in dairy cows but was not sustained over time.

## 1. Introduction

Anthropomorphic production of greenhouse gases is raising the temperature of the earth to levels that are threatening the sustainability of ecosystems on the planet [1]. Temperatures at regions on earth have risen since the 1960s by more than 1.2 °C and annual soil moisture is estimated to have declined by 20 to over 40% in the cropping regions of southern Australia [2]. Atmospheric concentrations of the major greenhouse gases, carbon dioxide and methane, have risen since 1950 from 350 to 410 ppm (28%) and 1100 to 1875 ppb (70%), respectively [3]. Methane is 28 times more potent than carbon dioxide as a greenhouse gas over 100 years and 80 times more potent over 10–20 years from release [4]. Enteric methane from the microbial fermentation of plant material by ruminant animals, primarily cattle, contributes 30% of methane released into the atmosphere, which is more than any other single source [5]. Enteric methane is the largest contributor (40%) to global greenhouse gas emissions from livestock supply chains [6], contributing 6% of total anthropogenic greenhouse gas emissions [7,8]. Ruminants also produce a substantial amount of carbon dioxide (CO_2_), with a methane:CO_2_ ratio of approximately 4:1 [9], making a total contribution of ruminants to anthropogenic greenhouse emissions of 8%. Consequently, there are demands from sections of the community for major reductions in the consumption of red meat to reduce the number of ruminants and amount of methane they release [10,11]. However, the Food and Agriculture Organization [12] projects that demand for red meat will continue to increase at the rate of around 1.5% per year to meet the growing population and rising living standards in developing countries. An alternative mitigation approach, rather than reducing red meat consumption or ruminant numbers, is to reduce methane emissions from ruminants.

Australia introduced the Emissions Reduction Fund (ERF) in 2016, where primary producers are paid for reducing greenhouse gas emissions provided an approved Method for methane reduction (or any other greenhouse gas) has government acceptance [13]. Methane emissions from ruminants reduce the energy from feed available for production, with losses to the animal of 3–12% of digested energy [14,15]. Reduction in ruminant methane emissions would decrease national greenhouse gas emissions and increase energy available to livestock for productivity.

As a forerunner to the Australian red meat industry establishing a goal to be carbon neutral by 2030 [15], researchers examined a wide range of potential methane mitigation strategies for ruminants in the National Livestock Methane Program (NLMP) coordinated by Meat and Livestock Australia [16]. The research incorporated numerous methane reduction strategies with some identified from previous reviews [17,18]. Results from this Australian research, complemented by published results from other sources, are used in this review to identify those strategies that are considered to have the highest potential for reducing methane emissions from ruminants.

## 2. Materials and Methods

Mitigation strategies examined in NLMP included: (i) the role of genetic selection; (ii) use of various feed supplements; (iii) potential role of anti-methanogenic forages; (iv) potential for methane emission reductions from a detailed understanding of rumen function; and (v) best pasture management practices. The potential role of vaccination against rumen *Archaea*, including peptide sequences, the commercially developed 3-nitrooxypropanol (3-NOP) [19] and biochar were not included in NLMP research, but were evaluated.

The evaluation involved assessment, for each potential mitigation strategy, of: (i) status of current knowledge; (ii) likely mitigation potential for individual animals and across Australia; (iii) impact on animal productivity; (iv) cost of and barriers to implementation; (v) chances of research success and its cost; (vi) likely delay before being implemented on production enterprises; and (vii) opportunity for ruminant enterprises to sell ‘carbon credits’. Steps (iv) to (vi) were based on subjective assessments made by 17 people comprising the NLMP investigators, members of the Investor Advisory Group, with a representative from each organization contributing funds to the program, and the NLMP management team [20].

These assessments have been used recently by Davison et al. 2020 [20] to determine the financial outcome for each mitigation strategy when applied to the Australian ruminant industry based on livestock types and numbers provided by Mayberry et al. [15]. Financial benefits from 2020 to 2030 were determined as the discounted additional income from changes in productivity and carbon credits based on the ERF price of AUD $16.14 per t of carbon dioxide equivalents, assuming methane to be 28 times more potent as a greenhouse gas than carbon dioxide. The discounted costs for purchasing and applying each strategy were then subtracted from the additional income to provide the net financial change, which was multiplied by the assumed adoption rate to provide an assessment of national monetary benefit to the Australian ruminant industries. Consequently, the methane mitigation strategies were ranked in declining order for likely financial benefit to the Australian cattle and sheep industries (Table 1).

## 3. Methane Mitigation Strategies

### 3.1. Red Macro Algae as a Feed Supplement

The red marine alga, *Asparagopsis taxiformis*, when collected in the filamentous tetrasporophyte phase, air-dried, ground and added at 2% of substrate organic matter, reduced methane emissions by up to 99% without depressing substrate digestibility or volatile fatty acid production in laboratory rumen fermentation cultures [21]. When dried red alga was fed to sheep for 75 days at five concentrations from 0 to 3% of organic matter, methane emissions declined linearly to over 80% at the highest dose [21]. Methane emissions did not increase over time, suggesting the rumen methanogen population did not adapt to the alga. A recent experiment with penned cattle showed that the inclusion of only 0.2% of feed organic matter as dried *Asparagopsis* reduced methane emissions by 98% and increased weight gain by 42%, without negative effects on feed intake or rumen function [22]. There were no changes in meat quality or detectable residues in the meat. Similar declines in methane production from 55% to 80% have been found in dairy cows fed 0.5% dry matter of bioactive *Asparagopsis taxiformis* [23]. Similarly, a decline in methane of 67% was observed when *Asparagopsis armata* was fed a 1% inclusion in a diet for lactating dairy cows with no residues observed in milk [24].

Stefenoni, et al. [23] observed a 5-fold increase in iodine and an 8-fold increase in the iodine and bromide concentration in milk from dairy cows fed total mixed ration diets containing 0.5% *Asparagopsis taxiformis* collected from Portugal compared with the milk from cows fed the control diet. Iodine concentrations are high in most seaweeds fed to animals. Although these higher-than-normal concentrations in milk could be detrimental to the health of young children, Stefenoni, et al. [23] believe this is not of major concern due to the normal mixing of milk from different farms. Stefenoni, et al. [23] did not detect significantly higher concentrations of bromoform in milk with 0.5% *Asparagopsis*, but it was detected in milk for the first 10 days after introduction at rates of 6.5% or more in a supplement to a dairy cow diet by Muizelaar, et al. [25].

Li, et al. [21] and Stefenoni, et al. [23] suggest that the palatability of diets containing *Asparagopsis* may be reduced, particularly at higher concentrations in the diet. Muizelaar, et al. [25] found dairy cows offered diet supplements with 6.5% or more *Asparagopsis* regularly refused feed and selected against these feeds. Previous studies have shown bromochloromethane can reduce methane emissions in cattle and goats by more than 90% without affecting feed intake [26,27,28]. The observed reduction in feed intake in several experiments with diets including *Asparagopsis* have been suggested to be due to the extremely high mineral content of the seaweed [24].

There are two reports of some animals showing changes in rumen papillae morphology and rumen ulceration with either prolonged (75 days) feeding [21] or feeding for shorter periods at 6.5% or greater of *Asparagopsis* in the diet supplement [25]. The diets used by Muizelaar, et al. [25] also contained large amounts of wheat that can cause rumen ulceration [29].

The *Asparagopsis* alga contains approximately 0.22 mg/g dry matter (DM) of 22 different halogenated metabolites but varies between strains and locations [30,31]. The concentrations of the bioactive compounds depend greatly on the drying procedures, with freeze drying being most effective [23]. The methanogenic activity of *Asparagopsis* declines over four months when incorporated into total mixed rations [23].

Halogenated methane analogues, such as bromochloromethane, inhibit methane production by reacting with reduced vitamin B12, which inhibits the cobamide-dependent enzyme methyl-coenzyme (CoM) reductase step in methanogenesis in a similar process as 3-nitrooxypropanol (3-NOP, see Section 3.2). The experiments reported have been conducted under pen conditions with diets containing grains or being high fiber, but pelleted [21]. The longest experiment was 90 days [22] and the potential impact of changes in vitamin B12 within the animal should be investigated over years if *Asparagopsis* is to be used in breeding cows.

In summary, the research conducted to date on *Asparagopsis* suggests that low concentrations of 0.5% or less of highly bioactive bromoform compounds in the diet can reduce methane emissions by up to 90% without detrimental effects on feed intake or product quality. Further research is needed to develop methods for decreasing the decline in bromoform activity when incorporated into diets. Research is also needed to show that the supplement can be fed to rangeland animals through lick-blocks or other methods to prove applicability for reducing methane emissions from rangeland and grazing breeding herds and flocks. If such applications are successful, *Asparagopsis* supplementation could be used across the whole of the Australian ruminant population.

*Asparagopsis* used in experiments has been collected manually in the wild. This process is expensive and not practical for commercial application. Research at James Cook University, Townsville, Australia, has measured variation in the concentrations and types of halogenated metabolites produced by algae due to genetic strains, sea water temperature and nutrient availability [31]. *Asparagopsis* can be cultured on ropes similar to the commercial culture of mussels and other macro-alga species. The alga can be grown in association with other aquaculture enterprises and could be valuable for reducing pollution from these industries. Due to the enormous potential of *Asparagopsis* to virtually eliminate methane emissions from ruminants and increase productivity, a company called Sea Forest has commenced commercial production of the alga in ponds in Tasmania [32]. A price for *Asparagopsis* being sufficiently low for practical methane mitigation in ruminant industries appears highly achievable. An ERF method needs to be developed if ruminant producers are to be paid for reductions in methane emissions. First practical application of *Asparagopsis* should occur within a year for feedlot and dairy animals, but will be longer for grazing stock.

### 3.2. 3-Nitrooxypropanol (3-NOP) as a Feed Supplement

3-nitrooxypropanol (3-NOP) and the ethyl variant, ethyl-3-nitrooxypropanol, were developed by DSM Nutrition Products Ltd. (Kaiseraugst, Switzerland) [33]. The compounds, bind to the active site of the enzyme methyl-coenzyme reductase that catalyses the last step in the reduction of CO_2_ to CH_4_ by the hydrogenotrophic methanogenic *Archaea* [34]. A potential advantage of 3-NOP over *Asparagopsis* is that it does not alter vitamin B12 metabolism. Laboratory and animal experiments show a significant reduction in methane when 3-NOP is added to the diet [19,35,36,37,38]. Methane emissions are reduced linearly in sheep, cattle and dairy cows as the proportion of 3-NOP is increased up to 2.8% of the diet [39].

Most animal experiments show a reduction in methane emissions of 8–30% except for the experiment of McGinn, et al. [40], where the reduction was 70%. The magnitude of the reduction in animal experiments is less than in fermentation assays where it frequently reaches 95%. The discrepancy between the types of experiments could result from the high volatility of the compound or a high rate of degradation within the rumen, which may be associated with the method of feeding in the animal experiments. With effective administration of the 3-NOP compound, methane emission reduction may exceed 40%.

There is no evidence 3-NOP reduces feed intake or digestibility of the diet and can be associated with an increase in productivity [39]. Hristov, et al. [19] reported that feeding 3-NOP increased body weight gain in lactating dairy cows, with no effect on milk production, but milk protein content increased. An increase in energy available to the animal would be expected [41] because hydrogen concentrations in the rumen are known to rise [39].

3-NOP may be available for all production situations because of the low dose rate required. An effective dose appears to be around 2% of the diet. The compound must be continually available in the rumen to be effective, which means it is best provided in total mixed rations to feedlot cattle and dairy cows. 3-NOP could be provided in lick-blocks or other supplements for grazing animals, but its effectiveness would depend on the frequency animals consumed these products.

DSM Nutrition Products is continuing research into feeding protocols for 3-NOP and is currently seeking registration for its use under the name Bovaer^®^ in ruminant diets in various countries around the world [33]. Bovaer is likely to be an important product for reducing enteric methane around the world, with its uptake depending on its price and whether there are payments for carbon reduction.

### 3.3. Manipulating the Rumen Microbial Population

Glucose from cellulose and starch consumed by ruminants is the main substrate fermented by rumen microorganisms in the rumen. Glucose is degraded by five competing biochemical pathways in the rumen, which produce different amounts of methane and convert different quantities of glucose energy to volatile fatty acids [42]. The most efficient pathway converts 93% of glucose energy to volatile fatty acids, with the production of no methane and high concentrations of propionate, whereas the least efficient pathway converts only 62% of glucose energy to volatile fatty acids and produces one mol of methane for each mol of glucose fermented with high concentrations of acetate. The other three pathways have an efficiency of conversion of glucose to volatile fatty acids from 72% to 86% and produce from 0.66 to 0.25, respectively, mol of methane per mol of glucose fermented.

Each of these fermentation pathways is predominantly performed by specific microbial species with only trace amounts of products from other pathways [20]. Methane mitigation strategies that substantially reduce methane emissions such as bromochloromethane, *Asparagopsis* or 3-NOP alter microbial populations to favour those that produce lower amounts of methane [43].

Abecia, et al. [43] and Meale, et al. [44] have demonstrated that if dams and their progeny are treated with a strong anti-methanogenic agent for up to two months after birth and the progeny remain isolated from other animals that have not been treated, the changes in microbial population that favours increased propionic acid production and reduced methane emissions can persist for up to a year after the treatment ceased. Abecia, et al. [43] showed methane emissions were 33% lower and live weight gain 18% higher four months after treatment for goats with the altered microbial population.

Meale, et al. [45] fed 3-NOP to one group of calves for 3 weeks post weaning and then ceased the supplement. Methane emissions were 10.4% lower in the treated group than the control at the end of the 3-week treatment. The difference in methane emissions between the previously treated group of animals and the control group was maintained at between 11.9% and 17.5% for up to 47 weeks after the treatment ceased. Distinct microbial populations remained evident between the treatment and control groups for the entire experiment.

The possibility generated from these experiments is for whole herds with desired rumen populations to be created and maintained through generations provided they are isolated from animals with different rumen populations [20]. The magnitude of the depression in methane emissions following cessation of the treatment is likely to be related to the size of the initial reduction. *Asparagopsis* supplements can reduce methane emissions by over 90% and, even allowing for some reversion in the microbial populations, a continuing reduction for long periods after cessation of the supplement of 30% would appear reasonable to assume. The strategy of manipulating rumen microbial populations should have practicality once the *Asparagopsis* or 3-NOP supplements are commercially available. However, further research is required to demonstrate the feasibility of maintaining isolated herds with modified rumen microbial populations and to demonstrate the magnitude of the reduction in methane emissions.

### 3.4. Tropical Legumes as Plantation Forage or Feed Supplements

*Leucaena* is a legume shrub grown in many tropical and subtropical regions of the world including the northern coastal environment of Australia, central Africa, Colombia, Mexico and Indonesia. *Leucaena* can be browsed and has the potential to be harvested, dried and included in rations for feedlot cattle or other ruminants. *Leucaena* contains high molecular weight condensed and hydrolysable tannins which bind to rumen microbes, reduce the population of *Archaea* and reduce methane formation [46]. However, *Leucaena* also contains the toxic amino acid mimosine and its breakdown product, 3-hydroxy-4-(1H)-piridon, which reduces growth rate, causes hair loss and ulcerations in the mouth and stomach [47]. These toxic products limit the acceptable intake of *Leucaena* to no more than 40% of the diet.

Cattle grazing *Leucaena* plantations with Rhodes grass or naturalized pasture showed a substantial increase in growth rate and a reduction in methane compared with cattle grazing pasture alone [48,49]. Similar improvements in cattle performance with reductions in methane emissions have been observed in Colombia [50] and Mexico [51,52].

Piñeiro-Vázquez, et al. [51] replaced a low-quality grass with up to 80% *Leucaena* in a pen experiment with cattle and found a linear fall to 60% of pretreatment enteric methane emissions. Browsing cattle on naturalized pasture appear to consume around 20% to 40% of their diet as *Leucaena* [53], which results in a reduction in methane emissions of 25% to 35%.

*Leucaena* may provide an alternative to silage or cotton seed in feedlot rations, but the effect of drying *Leucaena* on its ability to reduce methane emissions from ruminants is unknown. Based on feeding fresh *Leucaena*, a 20% inclusion in a feedlot diet would be expected to reduce methane emissions by 7% at the same feed intake [54].

Procedures for establishing *Leucaena* plantations are well known. The plant is generally not grazed until 18–24 months after establishment. The agronomy and costs for *Leucaena* planting and establishment are known. Developing an ERF method, so that producers can claim carbon credits when adopting a *Leucaena* feeding system, requires research to provide an accurate algorithm for predicting methane reduction and performance of cattle. The experiments could also provide near infrared (NIR) calibrations to estimate the proportion of *Leucaena* in the diet of an animal based on a scan of faeces [55]. Research is also needed to assess the effectiveness of dried *Leucaena*, followed by work with animals in feedlots.

Another tropical legume *Desmanthus* also contains condensed tannins that have been shown to reduce methane emissions in laboratory studies by 26% compared with a Rhodes grass control [56]. One experiment with cattle showed that including 31% *Desmanthus* cultivars in the diet replacing Rhodes grass resulted in a 10% decrease in methane emissions per kg of dry matter intake [57]. *Desmanthus* has an advantage over *Leucaena* because it can grow in semi-arid regions and has a potentially wider range of environments than *Leucaena* [58] where it can be incorporated into cattle production systems. However, further research is required to determine its potential for methane mitigation under grazing. As for *Leucaena*, an ERF method needs to be developed for *Desmanthus* if livestock producers are to obtain payment for methane mitigation.

### 3.5. Grape Marc as a Feed Supplement

Grape marc consists of skins, seeds, stalks and stems remaining after grapes have been pressed for wine. Grape marc contains condensed tannins, high concentrations of oils and tartaric acid, all with potential to reduce methane emissions in ruminants [59]. Tannins, particularly extractable tannin with shorter polymer chains and lower cis/trans ratio, were the most effective for reducing methane production by up to 50% and with little effect on total digestion in continuous 10-day laboratory rumen fermentation assays [60,61].

Grape marc has high fiber and low metabolisable energy (ME) content because of its high stalk and stem content [59]. Effectiveness of grape marc for reducing methane, without a negative impact on productivity, depends on the ME content of the marc relative to the diet ingredient it is replacing. Substituting 36% of a lucerne hay diet with marc pellets of similar ME content in feed for dairy cows in late lactation reduced methane emissions by approximately 20%, with little effect on milk yield [59]. However, when either white or red grape marc replaced fresh pasture with high ME content in diets for dairy cows in early lactation, milk yield was reduced by approximately 10%, while methane emissions were depressed by 15% with no differences between marc type [62]. The authors of the latter study conclude that the reduction on milk yield was related to the lower ME of grape marc, but the reduction in methane emissions was most likely due to the higher crude fat and lignin content reducing rumen fermentation than a direct effect of tannins. However, there is strong evidence from in vitro rumen fermentation studies that smaller sized extractable tannins reduce methane emissions without affecting the efficiency of fermentation by rumen microbes [62].

Replacing 30% of an oaten hay diet with grape marc of similar ME content reduced methane emissions from sheep by 10% without affecting animal productivity [63]. An Angus cattle feedlot experiment, where 20% of maize silage was replaced with marc, reduced emissions by approximately 10%, but growth rate was reduced by 25% due to lower energy intake [64].

These results suggest there is limited application for feeding grape marc to reduce methane emissions, without detrimental impacts on productivity. Grape marc sourced directly from the winery has no cost except for loading and transport. Costs increase with processing including ensiling, steam distilling, roller milling and drying for longer storage. Ensiling grape marc is the most practical method for storage on farms. Effective use of grape marc would be restricted to animal enterprises close to source because of the cost of transport. The best option for exploitation appears to be for sheep enterprises in southern Australia in proximity to vineyards during the summer-autumn feed-gap when pasture quality is poor or to sheep and cattle when feed supply is low.

### 3.6. Genetic Selection

Genetic variation in methane emission is inherent between individual ruminant animals [65]. Lower methane emissions may be manifest through a lower methane production for the same feed intake (i.e., residual methane production, RMP) and/or a lower methane associated with a lower feed intake at the same growth rate (i.e., low residual feed intake, RFI). The reduction in methane emissions in selected animals appears to be due to smaller rumen volumes, increased outflow rate of digesta and reduced fermentation in the rumen [66]. Heritability for methane emission, expressed as methane per unit of feed intake, is moderate (0.13–0.35) in cattle, sheep and dairy cattle [65,66,67,68,69]. Although heritability for methane emissions is moderate, the genetic variation between animals is small relative to other traits such as growth rate or milk production [70].

The impact of genetic selection on reducing methane emissions depends on the relative selection pressure placed on lowering methane emissions compared with other traits such as growth and reproduction. Changes to national methane emissions depends also on the speed the genes are passed through national herds. The potential for genetic improvement through direct selection for methane traits is limited at 0.2–0.4% per year, cumulating in a reduction of only 4–8% over 20 years for beef cattle and sheep [70]. The rate of genetic gain for methane mitigation is higher for dairy cattle because the widespread use of artificial insemination results in fewer sires used across the industry. Greater selection pressure can be applied in dairy sire selection than in the more extensive beef and sheep industries. Selection models for dairy cattle suggest methane reductions as large as 20–26% over 10 years are theoretically possible, but only at the expense of a 6 to 18% decrease in genetic gains for production traits [71]. Other models suggest the reduction in methane emissions from dairy cows would be only 3% over 10 years if there is genetic improvement in milk production traits [72].

Implementation of genetic selection for beef cattle would be relatively straight forward in Australia through the beef cattle genetic evaluation tool, BREEDPLAN. This procedure is used by cattle breed societies in Australia and has been modified to include RMP and RFI, which allows individual farm enterprises to include methane mitigation traits when predicting profit maximizing selection indices for breeding stock. The price for carbon credits, provided a payment method is available, influences selection pressure applied to methane mitigating traits [73], particularly when a price for carbon is higher, the selection pressure for lower methane emissions is increased relative to other traits influencing productivity [70].

Measuring methane from individual animal on-farm requires expensive equipment and intensive animal handling. Selection of low methane producing seed-stock for the beef cattle and sheep industries appears feasible only through genomic selection [74]. Although methane emissions have been measured on a limited number of Angus cattle and sheep in Australia, an estimated several thousand animals, representative of each industry, would be needed to develop a reliable dataset to utilize genomic selection for an individual breed. An exception is in the dairy industry, where genomic selection is established, few bulls dominate genetic improvement, and their genes are spread widely through artificial insemination. Superovulation and embryo transplant are also well established for dairy cows for rapid gene transfer. Methane emissions can be readily and relatively cheaply measured on a subset of their progeny.

Nevertheless, selection of low methane emitting sires by seed-stock breeders to produce commercial bulls or rams will be expensive and unlikely to be implemented. Similarly, the apparent conflict between reducing methane emissions in dairy cows and maintaining genetic gain in production traits, suggests genetic selection is not the most appropriate strategy.

### 3.7. Nitrate as a Feed Supplement

Non-protein nitrogen (NPN), typically urea, is fed to ruminants to increase microbial growth, feed digestibility, feed intake and productivity when crude protein concentration in the diet is less than about 60 g/kg DM [75]. However, when NPN is provided from nitrates, hydrogen is used in the conversion of nitrate to nitrite and then to ammonia. These nitrate reduction reactions have a lower free energy change than reactions utilizing hydrogen for methane production and therefore have a competitive advantage. Consequently, adding nitrate to ruminant diets reduces methane emissions, while providing NPN for microbial growth [76,77]. However, if the concentration of nitrite in the rumen rises and nitrite is absorbed into the blood, nitrite poisoning can occur through excess production of methaemoglobin in the blood [78]. Methaemoglobin reduces the oxygen carrying capacity of the blood and can result in death.

Numerous experiments have shown feeding nitrate can reduce methane emissions to a maximum of approximately 50% [78,79,80]. Theoretically, one gram of nitrate reduces methane production by 258 mg, but the average efficiency of hydrogen uptake is around 90% [80,81,82,83,84]. A rounded estimate is that 10 g nitrate/kg DM intake can reduce methane emissions by up to 10% [79]. The estimated maximum intake of nitrate without causing nitrite poisoning is approximately 20 g/day for beef cattle grazing low quality pasture [81], resulting in a methane emissions reduction of only 6.5%.

From evidence across many experiments, feed intake and growth rate of cattle are positively affected by nitrate feeding when rumen microbes respond to non-protein nitrogen [79]. However, research with feedlot cattle receiving 10 g nitrate/kg feed DM, which reduced methane emissions by 10%, also depressed feed intake by 10% [85,86]. In contrast, lactating dairy cows fed 20 g nitrate/kg of diet DM, reduced methane by about 15%, without an effect on feed intake or milk yield [77,80].

Experiments with sheep indicate similar responses to cattle when nitrates are included in either total mixed rations or supplements to lower quality forage diets [83,84,87]. However, there appears to be a consistent increase in wool growth up to 40% [82,86]. This increase is thought to be caused by nitric oxide formed from nitrite dilating blood vessels and increasing blood flow to the skin.

Nitrate is readily included in diets or lick-blocks as either calcium nitrate or ammonium nitrate as a full or partial replacement for urea. Integrity of lick-blocks is sometimes reduced because a greater proportion of the block is from the nitrate compound than from urea. The cost of nitrate compounds is higher than an equivalent amount of nitrogen from urea. The major concern with feeding nitrate for reducing methane emissions is the risk of nitrite poisoning, which is particularly dangerous when animals are subjected to exercise [81].

Encapsulation of calcium ammonium nitrate with a mixture of sesame gum and sesame oil cake enhanced the concentration of ammonia in an in vitro rumen fermentation system compared with the free nitrate, suggesting lower concentrations of nitrite [88]. Grazing steers supplemented with encapsulated nitrate at the rate of 47 g/100 kg live weight/day produced an average 10.6% less methane over 13 months compared with an equivalent amount of urea and grew slightly faster (*p* = 0.055), with no signs of nitrite poisoning [89].

Despite the potential benefits from encapsulation of nitrate sources slowing the rate of nitrite production, there is little practical value in using nitrates to reduce methane emissions from ruminants in grazing systems. The risk of nitrite poisoning remains a major concern and means the amount of nitrate offered would be conservative, with a small impact on methane mitigation. Although an ERF method for using nitrate to reduce enteric methane emissions has been approved by the Australian government, to date no enterprise had applied for a project using this practice.

### 3.8. Australian Shrubs or Plant Compounds as Feed Supplements

Several plant species, including the Australian Tar Bush shrub, *Eremophila glabra*, and the legume pasture plant, *Biserrula*, reduce methane emissions in laboratory rumen fermentation cultures and in sheep when compared with control diets [90,91,92]. Increasing amounts of *E. glabra* in a continuous rumen fermentation system reduced methane emissions linearly by up to 45% [92].

Introducing Australian native shrubs with anti-methanogenic properties into pastures is an effective method for reducing methane emissions and increasing productivity from sheep in south-west Australia [93]. The autumn feed-gap, with poor quality, senescent pasture is a major limitation to sheep productivity in the region. Traditionally, sheep are offered expensive supplementary grain during this period. Many Australian native shrubs grow well in the region and provide relatively high quality feed, with high protein content, particularly when consumed with senesced pasture [94].

Sheep offered native shrubs, with preserved inter-row pasture species, for 6–8 weeks in autumn, reduced methane emissions by 26%, increased growth rate from 69 to 142 g/day and eliminated the need for supplementary feeding [93]. Simulation modelling predicted the introduction of shrubs with inter-row pasture species increases whole farm profitability by an average of 24% when occupying an optimal 10% of the farm area [95,96].

Other bioactive compounds, extracted from native Australian *Melaleuca* and *Leptospermum* plants, reduce methane emissions by up to 97% in fermentation assays, but have not been tested in animals [97]. An estimate of the scale of reduction in methane emissions when bioactive compounds are provided as a supplement was obtained by comparing the reduction in methane emissions in the laboratory assay with reduction when the plants *Eremophila* and *Biserrula* were fed to sheep [97]. These comparisons suggest methane emissions may be reduced by approximately 25% if these bioactive compounds were fed to ruminants.

Laboratory studies suggest that the bioactive compounds could be included in diets at concentrations of 25–50 g/kg feed and reduce methane emissions, when offered as supplements or in lick-blocks. Although the cost of the bioactives is difficult to estimate, it is likely that they could be manufactured commercially. However, for the usefulness of these compounds to be assessed dose response experiments are required for the bioactive compounds to quantify their effects on methane over the longer term in ruminants. Initial experiments could be conducted with sheep in respiration chambers to determine responses over three months. If these experiments show significant and persistent reductions in methane, the effects of the compounds on feed intake and productivity will be needed as well as experiments with cattle. However, further research into these compounds appears unnecessary as the extraction procedures would be expensive and other supplements such as *Asparagopsis* species and 3-NOP (Bovaer^®^) are already proven enteric methane mitigants and are in commercial production.

### 3.9. Vaccination against Archaea

Vaccination against rumen *Archaea* has potential as a low-cost option to reduce methane emissions in sheep and cattle. The strategy is particularly attractive because it would require only one or two treatments in young animals for a lifetime effect and the practice is applicable to all ruminant production systems. Methane emissions were reduced by 8% in sheep vaccinated against a mixture of rumen methanogens [98]. A 20% reduction in methane emissions was regarded as highly probable when the ‘entire genetic repertoire’ of *Archaea* is examined to identify motifs common to all *Archaea*, but not to rumen bacteria [99]. Research in New Zealand is proceeding to identify possible antigens and develop a vaccine [100].

Attempts to reduce methane emissions through vaccination have returned varying results from 20% methane increase to 69% methane reduction with half the experiments being unsuccessful [101]. For a vaccine to be successful, high concentrations of anti-methanogenic *Archaea* IgG and IgA antibodies must be transferred from blood to saliva, which has proved difficult to achieve. Williams, et al. [102] found no effect of vaccination on methane emissions or on the number of methanogenic organisms within the rumen. These mixed results suggest that the likely success of a vaccination strategy for substantially reducing methane emissions from grazing ruminants is lower compared to other strategies.

### 3.10. Feeding Wheat to Grazing Dairy Cows

Feeding crushed wheat to dairy cows at a rate of approximately 9 kg across two feeds daily, reduced methane production by 30% to 40% with either freshly cut ryegrass pasture or chopped lucerne hay compared with crushed maize grain or pasture alone [103,104]. Milk yield was more than 20% higher on fresh pasture plus wheat. Moate, et al. [105] compared feeding daily with lucerne hay 10 kg of single rolled maize, or with single rolled wheat, or single rolled or double rolled barley replacing the maize. Methane emissions for the cows receiving the rolled wheat was 49% lower than for the maize diet, 73% lower than for the single rolled barley and 78% lower than for the double rolled barley. There was no effect of grain source on feed intake, but the energy and fat content of the milk from cows fed wheat was less than for the other diets.

However, when crushed wheat of small size with low starch content was fed at 9 kg/day with a long-cut lucerne hay compared with 9 kg/day of crushed maize fed, methane emissions were similar [63]. Milk yield was lower for the cows consuming crushed maize than for those consuming wheat, but methane emissions were similar between treatments. Although the composition or energy content of the wheat samples used in these experiments were not determined, the results suggest that normal, high starch wheat with rapid fermentation in the rumen is needed to substantially lower methane emissions.

Moate, et al. [105] showed a strong relationship between rumen pH and methane yield. The longer period of time rumen pH was below 6 the lower the methane yield. However, the positive effect on methane emissions of feeding high amounts of crushed wheat to dairy cows appears to dissipate over longer periods of feeding and the reduction in methane is at the cost of reduced milk fat [106].

Feeding 9 kg DM daily of wheat to dairy cows is not widely practiced for pasture-based systems in Australia, although some producers feed up to 12 kg DM wheat daily. The relative financial return from additional milk, but lower milk fat needs to be compared with the cost of feeding wheat. Providing wheat, with the correct starch composition and morphology, to cows consuming total mixed rations would be a simpler practice because it would replace other cereal grains and ingredients. Dose response curves are required to define relationships between wheat quality, daily wheat intake, methane emissions and milk yield for pasture based and total mixed ration feeding systems. This information would be required to develop an Emissions Reduction Fund method for dairy producers to claim carbon credits. Feeding high amounts of highly digestible wheat could lead to rumen acidosis unless managed carefully. Feeding large quantities of crushed wheat may only be practical in dairies using total mixed rations.

### 3.11. Biochar as a Feed Supplement

Biochar or biocarbon is produced from the partial pyrolysis of organic matter at high temperature to generate an extremely porous material with high surface area that is bioactive and binds organic compounds. The properties of biochar vary widely depending on the nature of the organic material and the conditions of partial pyrolysis [107].

Laboratory fermentation assays [108,109,110,111,112] and an early experiment with cattle [113] suggested biochar, as a feed supplement, may reduce methane emissions from ruminants. The experiment with young cattle offered a low-quality diet of dried cassava root chips and cassava foliage with 0.6% biochar increased growth rate by 25% to 140 g/day and reduced methane emissions by 22% without affecting feed intake.

Leng, et al. [110] showed the reduction in methane emissions in laboratory assays varied with the physical characteristics of biochar. Leng, et al. [113] postulated that the porous structure of biochar stimulates microbial colonization and biofilm formation, which enhances microbial growth and increases volatile fatty acids and protein supply to the animal. Methanogens are found on the outer surface of biofilms and are thought to remove hydrogen, which stimulates the digestion of cellulose and other feed compounds by maintaining a low hydrogen tension. The additional microbial growth and incorporation of hydrogen into microbes may be one reason for the decrease in methane production and increase in animal growth rate.

Despite the positive indications from in vitro experiments and theory, recent experiments with cattle fed from 0.5% to 3% of the diet as biochar made from pinewood [114] or whole pine trees [115] have failed to show significant declines in methane emissions. These latter results suggest that biochar is unlikely to have a significant impact on methane mitigation from ruminants.

### 3.12. Best Grazing Management Practices

A great deal of research has been conducted that has allowed development of on-farm practices that improve reproductive performance and efficiency of feed utilization. The major source of enteric methane emissions in Australia is the breeding female for cattle and sheep. A focus of management practice to reduce methane emissions is a reduction in the proportion of feed used to maintain this class of animal [18]. Improving reproductive performance and increasing growth rate of animals for sale reduces the total feed eaten by the herd used for animal maintenance and improves methane intensity (methane per unit of product sold). Reductions of approximately 20% have been indicated from improved grazing management systems for beef cattle [116]. The key red-meat advisory organization, Meat & Livestock Australia, sets out four methods for reducing methane emissions on cattle enterprises: (i) increase the ratio of live weight to age of animal in the herd; (ii) reduce the average age of the herd; (iii) reduce the proportion of unproductive animals in the herd; (iv) change the relative numbers of each livestock class. These changes can be implemented by improving grazing management to increase growth rate of progeny, weaning at younger ages and removing non-pregnant and non-producing cows. Adoption of best management practices across grazing enterprises in Australia was considered to improve productivity by approximately 20%, while reducing methane emissions by around 5% [16].

Factors limiting uptake of management practices known to increase the efficiency of feed use for productive functions are complex. Issues for consideration include relative advantages in productivity and profit, added complexity in management, compatibility with current practices and the risk associated with adopting the practice. Some best management practices may be simple to implement and others difficult. The cost of change within an individual enterprise can be significant in producer time and finances. Those changes that do not involve additional time or financial inputs, other than closer management of stock and resources, are most likely to be adopted. More use of the research into drivers and inhibitors of sustainable adoption of best practices may increase the speed and level of uptake of best practices on individual farms, regionally and nationally [117,118]. The strategy ‘Best Management Practices’ was not included in Table 1 because the wide range of practices potentially available t producers made allocation of costs impracticable.

## 4. Discussion

All herbivorous mammalian species, whether ruminants, foregut or hindgut fermenters emit methane and, except for equids, macropods and rabbits, produce on average 84 L/d per body mass raised to the power of 0.84 [9]. However, since the advent of farming around 8000 years ago, livestock numbers including ruminants, have increased in close association with the human population. In 2019, an estimated 43 Gt of carbon dioxide equivalents were produced annually from the world cattle population assuming 1.5 billion cattle [119], each producing 1500 kg of methane annually [20]. Thus, to contribute to anthropogenic reduction of greenhouse gases on global warming and help preserve ecosystems on the planet, it is crucial for ruminant enterprises to reduce enteric methane emissions. Most available strategies for reducing enteric methane emissions were reviewed to identify their mitigation potential, likely impact on animal productivity and practicality of adoption. Only four of the strategies: supplementation of diets with small amounts of the red seaweed *Asparagopsis* or the specifically designed chemical 3-NOP; long-term manipulation of the rumen microbial population; or feeding anti-methanogenic tropical legume shrubs appear to substantially reduce methane emissions and have potential practical application.

*Asparagopsis* and 3-NOP as feed supplements appear to have the greatest potential for reducing methane emissions by greater than 90% and more than 40%, respectively. Care must be taken to ensure the *Asparagopsis* has high bioavailability and is not fed at concentrations above 0.5% of the diet dry matter. These two products are theoretically available for ruminants in all livestock systems. They are undergoing commercialization and should be available for application, at first in intensive feeding systems, provided the costs are appropriate in relation to implementation and returns from any price on carbon dioxide equivalents. Further research is needed to develop appropriate processes for providing these supplements to grazing livestock through lick-blocks or other means.

There is also potential for intergenerational mitigation of methane emissions from ruminants through permanent manipulation of the rumen microbial populations. There is strong evidence [43,45] that rumen microbial populations can be changed for periods at least as long as one year to substantially reduce methane emissions. Such a strategy would be easy to apply, but careful management of herds to keep them isolated from other non-treated animals may incur practical difficulties. More research is needed to determine whether keeping herds isolated from other animals that have normal microbial populations is feasible and whether substantial levels of methane mitigation can be maintained over the long-term. Clauss et al. [9] speculate that the lower than herbivore average methane emissions from equids, macropods and rabbits is most likely caused by variations in the microbial populations fermenting forage.

Cattle browsing *Desmanthus* and *Leucaena* species resulted in methane emissions reductions of 10–20% and improved productivity of animals grazing naturalized pastures. However, their overall contribution to mitigating methane emissions will be relatively small because of restrictions imposed by the limited geographic regions over which these plants grow.

Many of the other strategies reviewed have a small impact on methane emissions from individual animals, apply to only a small proportion of the world’s herds, are a risk to animal health, are difficult to apply to production systems and demonstrate a negative commercial production cost/benefit analysis [20].

For example, genetic selection over 10 years, vaccination, grape marc, nitrate or biochar supplementation result in reductions in methane emissions of 10% or less. Best management practices resulted in small methane mitigation, but improvements in animal productivity, which makes them worthwhile for ruminant enterprises, though it can be difficult to sustain their application. Although, feeding large amounts daily of ground, high quality wheat reduced methane emissions in lactating dairy cows by around 35%, the persistence of the effect appears to diminish with time and there are risks to animal health through rumen acidosis.

The suggested priorities for application of methane mitigation strategies are similar to those listed by Beauchemin, et al. [8], except we have quantified impacts and placed a higher priority on programmed manipulation of the rumen microbial population and potential intergenerational persistence of lower methane emissions. However, most studies have investigated the impacts of individual methane mitigation strategies, with little assessment of at the effects of combined strategies.

The review did not consider variations in feed formulation through differing mixtures of forage and concentrates, addition of lipids, rumen modifiers such as ionophores, defaunation and additional phytocompounds including essential oils, saponins or flavonoids. Many of these methane mitigation strategies have been considered in recent publications [8,120] and are all regarded as having low methane mitigation potential. The value of combining mitigation strategies with different biochemical actions in the rumen, as suggested by Beauchemin, et al. [8], requires more than the few experiments that have been conducted to date. There is an opportunity to investigate such combinations, but if the supplementation of *Asparagopsis* and 3-NOP can be made practical and economic, persistence with strategies that have low methane mitigation potential appears unnecessary.

Uptake of mitigation strategies within Australia is encouraged through the ERF, which is a reverse auction process where projects bid for Australian Carbon Credit Units (ACCU) [121]. The current price is around AUD $16/ACCU, but only applies to ‘Methods’ which have been approved by Government. The Method must have specific parameters that can be measured to estimate the number of carbon credits achieved through methane mitigation from livestock. Currently, for ruminants in Australia, there are only Government endorsed methods for nitrate supplementation and beef herd management. The Beef Herd Management Method applies to projects that demonstrate reduced methane emissions as a result of new forages, new supplement use or new management methods that increase the growth rate of cattle to market weight at an earlier age or improve reproductive performance. Generic Methods are proposed in Australia for anti-methanogenic supplements in feedlot, dairy and grazing systems along with anti-methanogenic legumes, but are still dependent on further research.

Davison, et al. [20] estimated that by 2030 *Asparagopsis* supplements could be applied to 20% of Australian ruminants. With Australian greenhouse gas emissions in 2019 being 539 Mt carbon dioxide equivalents and ruminant enteric methane contributing 12%, a 20% reduction in enteric methane would result in 13 Mt less annually of carbon dioxide equivalents being released into the atmosphere by Australian farmers. That annual 13 Mt (in 2030) decrease in greenhouse gas emissions from adoption of *Asparagopsis* compares well with a reduction of 4 Mt in 2019 due to all renewable electricity production in the country and the 1.57 Mt reduction in enteric methane in the last 10 years (2009 to 2019) [122].

With over 1.5 billion cattle in the world [119] and each animal emitting around an average of 1500 kg carbon dioxide equivalents each year [20], adoption of *Asparagopsis* supplements at the same 20% rate by 2030, as assumed for Australia, would reduce global methane carbon dioxide equivalent emissions by half a gigaton annually. Application of methane reduction strategies have large potential for reducing global greenhouse gas emissions, while allowing the continued consumption of red meat.

Although this review is based predominantly on an analysis of the Australian ruminant industries, the assessments of the practical value of each methane mitigation strategy should be applicable to other countries and production systems. The feed supplements, *Asparagopsis* and 3-NOP, can be readily applied to any mixed ration formulation for feedlot and dairy animals, but methods need to be developed to ensure bioactivity of *Asparagopsis* can be maintained once it is incorporated into a diet. There is a greater challenge in utilizing these supplements in rangeland environments, where in Australia approximately 80% of enteric methane is produced from the breeding herd. The research being undertaken in Australia to investigate practical methods for supplementing rangeland animals would be applicable to many similar environments in other countries. Similarly, manipulation of rumen of the rumen microbial population should be applicable to production systems around the world where treated animals can remain isolated from untreated stock.

## 5. Conclusions

There is an urgent need for managers of ruminant enterprises to adopt strategies that reduce enteric methane emissions to avoid further increases in anthropomorphic associated greenhouse gas emissions. In Australia, this will most likely happen when alternate income sources from carbon projects can be realized and in turn this requires a substantive increase in research investment that results in new Methods that can be employed in carbon projects under the ERF. Potential methane mitigation strategies studied within an Australian research program and others were reviewed to identify their methane reduction potential, their impact on animal productivity and ease of application. The review concluded that only supplements of the red seaweed, *Asparagopsis*, and 3-NOP have potential for cost-effective practical application and can reduce methane emissions by 90% and 40+%, respectively, with likely increases in animal productivity and without negative effects on animal health or product quality, provided *Asparagopsis* does not exceed 0.5% of the diet. Both products are undergoing commercialization. There is also potential for intergenerational mitigation of methane emissions through permanent manipulation of the rumen microbial populations and keeping these animals isolated from other non-treated animals. However, further research is needed to determine the long-term feasibility of the strategy and magnitude of methane mitigation. An assumed annual reduction of enteric methane emissions of 20% by 2030 would reduce greenhouse gas emissions by half a gigaton each year.

## Figures and Tables

**Table 1 animals-11-00951-t001:** Potential benefit to Australian ruminant industries from adopting different methane mitigation strategies.

Methane Mitigation Strategy	Animal Methane Mitigation Potential	National Methane Mitigation Potential ^a^	Productivity Gain	National Methane Mitigation Potential ^a^	Proportion of Beef Herd Implicated	Expected Adoption	Benefit to Australian Industry	Time to Application
	(%)	(Mt/y)	(%)	(% Total)	(%)	(%)	AUD/y	y
*Asparagopsis*	90	42.91	20	66.22	100	20	$2695.82 M	1–5 ^b^
3-NOP	40	4.77	3	7.36	100	5	**	1–5
Microbe manipulation	30	11.44	20	17.66	80	20	$979.25 M	1
*Desmanthus*	15	2.86	15	4.41	40	20	$195.79 M	Now
*Leucaena*	18	1.72	20	2.65	20	20	$29.69 M	Now
Grape marc	10	0.24	0	0.37	2	50	$12.14 M	Now
Genetics	7	0.50	0	0.77	100	3	$4.36 M	Now
Nitrate	5	0.07	0	0.11	4	5	−$2.61 M	Now
Shrubs	4	0.06	5	0.09	5	5	−$566.56 M	Now
*Biserrula*	16	0.11	−15	0.18	3	10	−$1378.22 M	Now
Vaccination	5	0.12	2	0.18	100	0	0	
Wheat feeding dairy	35	0.00	10	0	2	0	0	
Biochar	0	0.00	0	0	0	0	0	
Total		64.8		100				

^a^ Based on 64.8 Mt/y total Australian enteric methane CO_2_ equivalents (see Section 4) and animal types and numbers from Mayberry, et al. [15]; ^b^ 1 year for feedlot and dairy and up to 5 years for grazing animals; ** No financial calculations were made for 3-NOP because it is a commercial product not yet released.

## Data Availability

No new data were created or analyzed in this study. Data sharing is not applicable to this article.

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
