# Peer review of "Methane Emissions from Ruminants in Australia: Mitigation Potential and Applicability of Mitigation Strategies"

_animals, 2021, doi:10.3390/ani11040951_

Round 1
Reviewer 1 Report
I'm satisfied with your revisions, and appreciate that you took my suggestions to heart.
Author Response
Pleased Reviewer 1 was satisfied with our responses.
Reviewer 2 Report
The reviewer appreciates their sincere efforts to address all concerns by reviewers and the editor
Author Response
Pleased the reviewer was satisfied with our response.
Reviewer 3 Report
The article is interesting and well written, and it gives a useful description of strategies to mitigate ruminants’ methane emissions in Australia. Thank you for having considered my suggestions.
Author Response
Pleased the reviewer was satisfied with our response.
This manuscript is a resubmission of an earlier submission. The following is a list of the peer review reports and author responses from that submission.
Round 1
Reviewer 1 Report
General
I believe your review is in good shape, so my comments focus on efforts to clarify your main points.
I think the recent review of Clauss et al. 2020 (Animal 14:S1, pp s115-s123 ) (attached to this review) has information that could make your review more interesting and germane to a larger audience, by putting ruminants in perspective with other herbivores,
If I’m correct in my calculations from the Clasuss review, Total GHG equivalent emissions from ruminants are 4:1 methane:CO2, which shows that the CO2 part of ruminant GHG emissions is not trivial.
I suggest a look at Ripple et al. 2014 " Ruminants, climate change and
climate policy" (Nature Climate Change volume 4, pages 2–5)  for their approach to the topic, and perhaps some ideas to enhance your points.
Line Comment
13 suggest the more accurate term ‘eructate’ or ‘release methane in the course of rumination’; ruminants don’t ‘burp’ (at least you didn’t say ‘cow farts’)
15 add ‘methane’ in front of ‘source’, to clarify that it is not percent of GHG.
47 Somewhere in Introduction I recommend clarification of methane and GHG equivalents. From your citations it appears that the livestock sector is responsible for approximately 14.5 % of anthropogenic GHG sources after accounting for the 80 X potency of CH4 in GHG equivalents (FAO, 2013), and 44% of the livestock sector comes from ruminant methane, so ruminant methane is 6.5% of global anthropogenic GHG emissions. I might be pertinent to point out that Gerber et al (2013) and Beauchemin (2020), and you in this review are making calculations based on 2004 data in the FAO 2013 report. Things might have changed in the past 17 years.
91 Consider using a rating system similar to that used by Gerber et al in their review, e.g. ‘low’,, ‘medium’, High. Their criteria are reasonable and could be adapted to your review.
102 ‘smaller’, not ‘small’’ Heritability of reproductive traits is ‘small’.
135 how about superovulation, embryo transplant to speed up the process.
145 clarify that this is extent of digestion, perhaps not within a physiologically feasible time frame (slower rate of fermentation), especially if these are closed fermentations over several days.
408 Is this on a DM basis? If not it should be; feedlot diets are formulated on a DM basis.

Reviewer 2 Report
Comments to the author
In this review manuscript authors described general understandings of currently proposed approaches for the mitigation against ruminant methane emission. Compared to already saturated amount of review papers in this research area, their work is almost a simple collection of reports and review papers from the beginning to the end, the reviewer does not think it so remarkable and significant at this stage. This drawback would also relate to the title, indeed they dealt with some statistical data and research examples conducted in Australia but it does not much reflect what is the particular mode in the country. Authors should better address how different would the country take suitable measure among various options for the mitigation, and compared to other countries like Europe and the US. Another concern is obvious that all the things consisted of text only, no table, no figure. This makes subscribers be tired for reading so long pages, which also contains much room for shrinking.
Eventually the reviewer like to suggest how it would be amended:
1) Consider to cut off the redundant parts that have been already addressed in their previous literature (ref # 17), section2.5, 2.6, and 2.8. Dependence to this publication also exhibits some unbalance of topic to beef cattle, while more attention must be paid to dairy cattle given comparing the GHG impact. Instead it would provide much impact on subscribers if they add information about how much extent cattle farms are ACTUALLY paying their efforts to cut GHG gases in their operations by introducing practical measures described here.
2) Also it would be appreciated if authors address the sections 2.11 and 2.12, in which the argument is too generic to be involved in this article, as these topics have been already done in previous literature. 
Minor comments
- Abstract is just a general summary of current situation in the world, therefore pl rewrite to help subscribers seize the main body of this paper.
- Many of reference information are depending on wrong formatting, most of them are either Headline Style of Title or full spelling of journal title. Correct.     
Reviewer 3 Report
Interesting and appealing article, it really gives a noteworthy detailed description of ruminants’ methane emissions mitigation strategies in Australia. However, I would suggest a minor revision in the section structure:
"Introduction" adequately describes the research justifications and aims but it contains also what was examined by researchers and what was incorporated in the research (lines 72-75), along with a more detailed description of what such examined mitigation strategies included (lines 76-81). Moreover, it is described the assessment involved in the evaluation for each potential mitigation strategy (lines 82-90).
I suggest rearranging this part from the “Introduction” section as a “Materials and methods” section to better separate it from the aims and justifications of the research.
